# Characterization of Unidirectional Replication Forks in the Mouse Genome

**DOI:** 10.3390/ijms24119611

**Published:** 2023-06-01

**Authors:** Avital Zerbib, Itamar Simon

**Affiliations:** Department of Microbiology and Molecular Genetics, IMRIC, Faculty of Medicine, Hebrew University of Jerusalem, Jerusalem 91120, Israel; avital.zerbib@mail.huji.ac.il

**Keywords:** origin of replication, G4 quadruplex, ori-SSDS, replication fork direction, bidirectional replication, replication fork barrier, unidirectional replication

## Abstract

Origins of replication are genomic regions in which replication initiates in a bidirectional manner. Recently, a new methodology (origin-derived single-stranded DNA sequencing; ori-SSDS) was developed that allows the detection of replication initiation in a strand-specific manner. Reanalysis of the strand-specific data revealed that 18–33% of the peaks are non-symmetrical, suggesting a single direction of replication. Analysis of replication fork direction data revealed that these are origins of replication in which the replication is paused in one of the directions, probably due to the existence of a replication fork barrier. Analysis of the unidirectional origins revealed a preference of G4 quadruplexes for the blocked leading strand. Taken together, our analysis identified hundreds of genomic locations in which the replication initiates only in one direction, and suggests that G4 quadruplexes may serve as replication fork barriers in such places.

## 1. Introduction

DNA replication is initiated at origins of replication. While the bacterial genome is replicated from a single origin, eukaryotic genomes are replicated by many origins of replication. Replication is initiated at each origin and proceeds bidirectionally. Mammalian genomes are replicated by 30,000–50,000 origins, each activated at a characteristic time during the S phase [1,2]. The origins are not determined by a consensus sequence, but rather by contextual cues [3], including vicinity to TSSs (transcription strat sites), CpG islands, nucleosome-free regions, G4 quadruplexes, and accessible chromatin [2,4].

One of the big challenges in the replication field is the identification of genomic locations from which replication initiates. Several methodologies have been developed, including a method based on the isolation of short nascent strand (SNS) DNA, which is protected by an RNA primer, a combination only expected at origins of replication. Finding enrichment of such SNS in a particular genomic location is indicative of an active origin [5].

Recently, the SNS methodology for origin identification was further improved by the addition of a strand-specific sequencing step to the technique (Appendix A). The new method is called origin-derived single-stranded DNA sequencing (ori-SSDS). The advantage of this methodology is that it filters the noise out of the SNS data since it uses an additional criterion for active origins—their bidirectionality. For example, the ori-SSDS technique was recently used to accurately identify origins that are active during meiotic replication [6].

While most of the identified SNS peaks appear on both strands and thus represent bidirectional origins, the ori-SSDS method identifies thousands of peaks that are present in only one of the strands. What are those non-symmetrical peaks? One possibility is that they represent genomic regions in which the replication fork collapses and the non-symmetrical peaks are a consequence of restarting the replication processes, possibly through priming by the PrimPol protein [7,8,9]. Alternatively, these could be origins of replication that fire only in one direction due to replication fork barriers.

Replication fork barriers (RFB) are genomic regions that cause replication fork stalling or pausing, frequently due to proteins bound to them [10]. In *E. coli*, the Tus protein blocks replication at the Ter sites in an orientation-dependent manner [11]. Similarly, in budding yeast, Fob1 is bound to the RFB in the rDNA region and blocks replication in one direction, ensuring coordination between replication and transcription [12]. In contrast to RFB, in which the fork stalls, there are genomic regions that cause transient pausing of the replication forks. For example, the yeast centromeres and tRNA regions contain sites in which replication is transiently paused and then resumes again [13]. Replication fork stalling and pausing has also been reported in mammalian cells [14]. The physiological role of fork stalling and pausing is still poorly understood; however, it has been suggested to be involved in multiple physiological processes, including recombination, preventing replication–transcription collisions, kinetochore complex assembly, yeast mating-type switching, replication termination and telomere replication [14].

G4 quadruplexes are non-canonical DNA structures that can be formed in certain DNA sequences [15]. They are enriched around origins of replication and were found to be associated with origins in the ori-SSDS data as well [1,6]. Although this association was initially suspected to be an artifact of the SNS method stemming from the inhibitory effect of the G4 quadruplexes on the lambda exonuclease, it was later proven to be functionally associated with origins [16]. G4 quadruplexes can also interfere with the replication process. Replication forks that encounter a G4 structure are paused. The resolution of the G4 structure is done by specific helicases that are specially recruited to the paused replication fork [15,17], including FANCJ [18,19,20], Mgs1 and Pif1 [21], and Timeless and DDX11 [22].

Here we explore the nature of the non-symmetrical peaks by analyzing separately those that are found on the Watson and on the Crick strands and compare them to the bidirectional peaks. To our surprise, all three types of peak exhibit characteristics of origins of replication. Nevertheless, they differ in their association with G4 quadruplex sequences. We found that in contrast to bidirectional origins, the unidirectional peaks have a non-symmetrical association with G4-forming sequences. Those sequences are enriched for the leading strand on the side in which the replication is stopped, suggesting that the G4 on the leading strand may cause the replication fork to pause, resulting in unidirectional origins. These results are important, since they expand the concept of unidirectional origins to mammals.

## 2. Results

### 2.1. Single-Stranded Ori-SSDS Peaks Are Origin of Replication

Classical origins of replication promote bidirectional replication and therefore appear in ori-SSDS data as peaks in both strands—the Crick strand peak is 5′ and the Watson strand peak is 3′ of the origins (Figure 1A; right). What are the non-symmetrical unidirectional peaks? We first hypothesized that they represent regions in the genome in which a fork collapsed and then restarted, producing a short nascent leading strand only in one direction. In such regions, we would expect to see peaks either on the Crick or on the Watson strand, depending on the direction of the restarting fork (Figure 1A, left).

In order to explore the nature of the unidirectional peaks, we followed Pratto et al. [6], and for each ori-SSDS peak defined whether it had reads on both strands (origins) or only in Watson or Crick strands. We divided the data into four quadrants that differ in the fraction of reads mapped to the Watson and Crick strands. While quadrants 2 and 3 are characterized by reads mainly in the Crick or Watson strands, respectively, quadrant 4 contains peaks in which reads are mapped to the Crick strand in the left part of the peak and to the Watson strand in the right parts of it and thus represents classical origins (Figure 1B and Appendix A).

Next, we wanted to explore the fork direction of each of those peaks. To this end, we used mouse ES cell OK-seq data, which mapped fork directionality along the genome by mapping which strand was enriched with Okazaki fragments [23]. We expected to see a change in fork directionality at origins, whereas in the Crick-only and Watson-only peaks (CO and WO, respectively), we expected to see the fork moving in a single direction. To our surprise, all three types of peak behaved identically, with a switch in fork direction in the middle of the peak, a characteristic of origins of replication (Figure 1C). This suggests that all the peaks represent bona fide origins, most of them (81% and 66% in ES and testis, respectively) bidirectional, yet a big portion of them (18% and 33% in ES and testis, respectively) unidirectional. Those unidirectional origins reflect cases in which one of the replication forks is stalled immediately after replication initiation and probably resumes replication later (see Discussion).

### 2.2. Characterization of the Unidirectional Origins

Next, we characterized the genomic distribution of the three types of origins in both ES cells and testis. To this end, we used the ChIP seeker tool [24] and analyzed the association of each type of origin with genes, promoters and intergenic regions (Figure 2A and Appendix A). We discovered that bidirectional origins are enriched in promoters and depleted from genes and intergenic regions, whereas the unidirectional origins are enriched and depleted to a lesser extent in promoter and intergenic regions, respectively, and are neither enriched nor depleted in genes. No association was found between the three types of peak and repetitive sequences (Figure 2B and Appendix A).

The difference in the association of the different type of origins with promoters and with genes prompted us to explore the association of the atypical origins with other genomic features known to be associated with origins. Similarly to what was shown for classical origins [25], unidirectional origins are also enriched in high-GC-content regions (Figure 2C and Appendix A), CpG islands (Figure 2D and Appendix A) and sequences that can form G4 quadruplexes in vitro [26] (Figure 2E and Appendix A). Here, we found also that the association of these genomic features with bidirectional origins is significantly stronger than the association with the unidirectional origins.

Next, we explored the association of the three types of peak with epigenetic features, including gene expression, chromatin accessibility, and replication timing. We used RNA-seq data of mouse ES cells [27] to analyze the association of the peaks with transcription levels and limited our analysis to the top 8103 expressed genes (RPKM > 5). We found that although both types of origins are not enriched in genes as a rule (Figure 2A), they are enriched in the group of highly expressed genes, with a stronger enrichment for bidirectional origins (Figure 3A). Similar results were observed for the association with open chromatin. Analysis of ATAC-seq data from ES cells [28] revealed that both types of origins are enriched in accessible genomic regions, with stronger association for bidirectional origins (Figure 3B). Finally, as origins are enriched in early-replicating genomic regions [25], we compared the distribution of replication timing of the entire genome in ES cells [29] and testis [30] with the distribution of the replication timing of the three categories of peak. We found that the distribution of the replication timing of all three types of peak are significantly different to the genomic distribution of replication timing (*p* value~0; Kolmogorov–Smirnov (KS) test) and are highly enriched in early-replicating regions (Figure 3C and Appendix A).

Taken together, we found that the unidirectional origins are associated to similar genomic and epigenomic features as the bidirectional origins. Nevertheless, their association with promoters, CpG islands and with open chromatin and expressed genes is less pronounced.

### 2.3. What Is Unique in Unidirectional Origins?

The results presented so far may suggest that the unidirectional origins are weak origins that were not detected on both strands, explaining why they are less enriched for most known characteristics of bona fide origins. Alternatively, they may be strong origins that fire only in one direction. In order to address this possibility, we searched for specific characteristics of these origins that may explain their unidirectionality.

It has been suggested that G4 quadruplexes may interfere with replication fork progression in both the leading and the lagging strand [15]. Our analysis allows us to test this, since the Watson-only and Crick-only peaks may represent origins in which the leading strand progression was in only one direction (and therefore lack a peak in the other strand). Such peaks may be a consequence of fork pausing in the other strand due to G4 structures. Thus, looking at the strandness of each G4 structure in the different types of peak may reveal the effect of G4 structures on fork progression. To this end, we explored the strandness of the G4 sequences in the three types of peak, limiting our analysis to peaks that contain G4 in only one of the strands (4216 and 2033 in ES and testis, respectively). We found that the CO peaks were highly enriched with G4 quadruplex structures on the Watson strand, whereas the WO peaks showed exactly the opposite bias (G4 enrichment in the Crick strand). The bidirectional origins did not exhibit any bias. The same biases were found both in ES cells and in testis (Figure 4A and Appendix A). These results suggest that the WO and CO peaks are genuine unidirectional origins. We propose that the G4 quadruplex on the leading strand perturbs the progression of the leading strand of the replication fork, forming unidirectional origins. Furthermore, expanding our analysis to the 2 kb surrounding the center of the peaks and counting all the G4 quadruplexes in each strand in both directions of the peak revealed the same strong bias toward G4 formation on the opposite strand of the peak for the unidirectional origins, with no strand specificity for the bidirectional origins (Figure 4B and Appendix A).

## 3. Discussion

Origins of replication are generally considered to be genomic regions in which bidirectional replication starts [1]. However, we wondered whether there were origins in the mammalian genome that challenged this definition. In order to address this question, we took advantage of a recently published dataset that mapped origins in a strand-specific manner using a version of the SNS methodology [6]. We found a significant number of origins that fire exclusively from either the Watson or the Crick strands. Those unique origins share most of the features of regular origins, including high GC content, enrichment on promoters, CpG islands, G4 quadruplexes, accessible regions, transcribed regions and early-replicating regions. However, the enrichment in these genomic regions was less profound.

Analysis of replication fork direction (through OK-seq data) raised the possibility that the non-symmetrical peaks are origins of replication that somehow fire only in one direction. Alternatively, such non-symmetrical peaks could stem from places in which the fork collapses and restarts by the activity of the PrimPol primase [7,8,9], for example, in the vicinity of G4 quadruplexes [31]. The failure to find such cases of fork restart could be a sign that such fork restarts are sporadic and occur in each cell in a different location and thus cannot be detected using bulk measurements. Alternatively, even systematic fork restarts will not be detected by SNS-based methodology since it is based on the isolation of RNA-protected short nascent strands of DNA, and the PrimPol primase can use both dNTPs and NTPs for priming DNA synthesis [8].

We found that 18–33% of the origins fire in one direction (Figure 1B), yet OK-seq analysis revealed that in all of them, the forks proceed on both sides of the origins (Figure 1C). This is surprising, since one does not expect to see replication in the direction of the blockage. The most likely explanation for this apparent contradiction is that the fork blocking is transient and the blocked fork restarts later, probably by the activity of the PrimPol polymerase [7,8,9], and the replication proceeds in the same direction. Further experiments in which OK-seq is performed in cells lacking PrimPol activity are needed to shed further light on this issue. An alternative explanation is that even without fork collapse, the replication resumes after a delay and cannot be detected as an ori-SSDS peak. The replication delay allows the ligation of a few Okazaki fragments prior to the synthesis of the leading strand, and thus when it is synthesized it is immediately too long to be detected as a SNS due to its ligation to a relatively long lagging strand. 

Our results are based solely on the analysis of the ori-SSDS data and we interpret them assuming that ori-SSDS non-symmetrical peaks represent cases in which the replication was paused in one direction and therefore the short nascent strands were isolated only in one strand. Alternatively, the lack of peaks may be a consequence of selective loss of the RNA primer in one of the strands or even a sign of quicker synthesis in one of the strands causing a smaller fraction of the cells to harbor a nascent strand at the isolated length. However, the finding of G4 quadruplexes on the absent leading strand (Figure 4) strongly supports our interpretation of the data. Further research in which fork progression is studied directly is needed in order to prove the existence of such unidirectional origins.

Our analyses revealed that there are two types of origins—the classical bidirectional origins and origins that fire mainly in one direction. We found that the unidirectional origins are enriched with G4 quadruplexes on the Watson strand for CO origins and on the Crick strand for the WO origins. These results are consistent with the assumption that G4 structures impede replication mainly on the leading strand, and thus, if they appear on the Watson strand, we detect leading strand synthesis only on the Crick strand and vice versa. Interestingly, classical bidirectional origins are characterized by balanced amount of G4 in both strands (Figure 4), with a significant bias of the G4 quadruples toward the lagging strand (Appendix A), supporting the idea that G4 quadruplexes interfere mainly with leading strand synthesis (Figure 5).

G4 quadruplexes have been associated with the replication processes both in the context of origin firing and as a factor that can interfere with replication fork progress (review in [15]). Although G4 quadruplexes can form on both strands, lagging strand synthesis seems to be more tolerant to this structure given the continuous ability to reprime the lagging strand. On the other hand, plenty of evidence suggests that G4 quadruplexes on the leading strand may interfere with DNA replication, especially when they are stabilized or in the absence of specialized helicases [31,32,33,34]. Our results are consistent with these observations, and to the best of our knowledge are the first example of interference of G4 quadruplexes with replication under normal unperturbed conditions.

For G4 locations, we used the G4-seq data that measures the potential of a sequence to form G4 quadruplexes in vitro [26], yet only quadruplexes that are stable in vivo may interfere with the replication process. The factors that affect quadruplex formation and resolution in vivo are not fully understood [15] and further research is needed in order to confirm the involvement of G4 quadruplexes in the determination of fork direction in vivo. Interestingly, using the recently published G4 CUT&Tag data [35], which represents the existence of a G4 structure in vivo, revealed that the three types of peak are enriched with G4 structures (Appendix A). These results are similar to the in vitro results presented in Figure 2E, yet since the CUT&Tag data does not contain information about the strands, it cannot distinguish between the bidirectional and unidirectional origins.

Another possible interference with replication fork progression is the transcription machinery. It has been suggested that head-to-head collisions between the replication and transcription machineries cause replication fork collapse. Indeed, we found a small but significant enrichment for transcription of the opposite strand in the unidirectional origins, suggesting that head-to-head collisions are indeed deleterious for leading fork progression (Appendix A).

Some proteins that bind to the DNA interfere with replication fork progression and in some cases form a replication fork barrier (RFB). This has been demonstrated in yeast, in which due to an RFB, the replication in the rDNA region is unidirectional (reviewed in [36]). Such barriers can also be introduced into mammalian genomes by the integration of the *LacO* sequences and expression of the lacI gene [37]. Similarly, induction of the Tus-Ter bacterial system in mammalian cells blocks replication forks [38]. Our results expand the concept of RFBs and suggest the existence of hundreds of genomic loci in which the replication initiates in a single direction. Further research is required to confirm our results experimentally and to decipher the actual barriers in each case.

## 4. Conclusions

Our analysis of the non-symmetric ori-SSDS peaks revealed numerous instances of unidirectional origins. These unique origins are similar in their genetic and epigenetic characters to the classical bidirectional origins, and probably represent cases in which the replication forks were paused in one direction. Our bioinformatics analyses revealed the involvement of leading strand G4 quadruplexes in delaying fork progression. Further research is needed to confirm these results in vivo, to identify additional factors that can serve as replication fork barriers in mammalian cells, and to decipher the functional importance of such unidirectional replication.

## 5. Methods

A schematic representation of all the analyses performed in the research is shown in Appendix A.

### 5.1. Definition of CO, WO, and Origin Peaks

Since we expect a switch from Watson to Crick strand at the middle of the origins, we followed [6] and divided each peak at its middle and counted the number of reads that mapped to each strand in the right and left sides of the peaks. The fraction of each strand in the left and right parts of the peaks was calculated using the following equations and is plotted in Figure 1B.
Left fraction = left (Crick or Watson)/(left_Crick + left_Watson),
Right fraction = right (Crick or Watson)/(right_Crick + right_Watson)

Crick only (CO) peaks were defined as peaks with Crick fraction both in the left and the right parts > 0.5; Watson only (WO) peaks were defined as peaks with Crick fraction in both the left and the right parts < 0.5; and origins were defined as peaks with Crick fraction > 0.5 in the right part of the peaks and <0.5 in the left part of the peaks.

### 5.2. Randomize Regions

For the asymmetric and symmetric peak files (CO, WO and ORC), we created a corresponding random file that contains random genomic regions with the same amount and size as the peaks. Those regions were chosen from the mappable parts of the mm10 genome using the randomizeRegions R program (randomizeRegions(A, genome = “mm10,” mask = NULL, allow.overlaps = TRUE, per.chromosome = FALSE,…)).

### 5.3. Okazaki Sequencing Data Processing

For Ok-seq data, we used the data published in [23]. The ES cell OK-seq data were downloaded from ENA (accession number SRR7535256). The data were trimmed using the cutadapt tool (cutadapt -a AGATCGGAAGAGCACACGTCTGAACTCCAGTCA -q 20–minimum-length 40) and mapped with BWA-0.7.12 (https://github.com/kaist-ina/BWA-MEME/ (accessed on 19 February 2023) to the mouse genome (mm10).

Reads that were not uniquely mapped were removed using samtools (samtools view -hbS -q 30 -F 0 × 100) and duplicate alignments were removed with Picard Tools MarkDuplicates (http://broadinstitute.github.io/picard/ (accessed on 19 February 2023)). Only reads with mapping quality > 30 were used in the analysis. The reads mapped to each strand were counted and binned in 1 kb windows by overlap with a file of 1 kb windows by Bedtools makewindows as described in [39].

### 5.4. Replication Fork Directionally

RFD was computed using the equation RFD = (R − F)/(F + R) as described [23]. F and R corresponds to the number of mapped reads to forward (Watson) and reverse (Crick) strands, respectively. For origin identification, we followed [6] and plotted the average RFD values (50 kb windows) for 40 windows centered at the ori-SSDS peaks from each category.

### 5.5. Genomics Feature Analysis

The overlap between the three types of peak with known genomic features were performed with the ChIP seeker tool (10.18129/B9.bioc.ChIPseeker). In Figure 2A, we used the following definitions: promoter = within 1Kb of the TSS of a gene; gene = the 5′ UTR, 3′ UTR, 1st exon, other exon, 1st intron and other intron; intergenic = distal intergenic.

Data on repeats and CpG islands were downloaded from UCSC (https://genome.ucsc.edu (accessed on 30 May 2022)) using the RepeatMasker and CpG island tracks, respectively.

For GC content analysis, we used *Bedtools nuc* [39]. We extracted from the output information the GC content column.

G4 quadruplex data were downloaded from [26], which represents the potential of the sequence to form G4 quadruplexes in vitro separately for the Watson and the Crick strands.

In all cases, the overlap was determined between the peak and the genomic feature, requiring a minimum overlap of a single base.

### 5.6. Epigenomics Feature Analyses

For RNA-seq analyses in ES cells, we used data (GSE135557) from [27]. We normalized the CPM data for gene length and turned them to RPKM. We defined expressed genes as genes with RPKM > 5 (*n* = 8103). For strand-specific analyses, we used information from UCSC genome browser about the transcribed strand.

For ES cell ATAC-seq analysis, we used data (GSM1563569) from [28].

For replication timing analysis, we used testis data from [30] and ES data from [29].

The transcription direction and the RT were assigned to the middle of the replication peaks.

### 5.7. Statistical Analyses

For comparisons of distribution, we used two-tailed *t*-tests (Figure 2C and Figure 3B) and Kolmogorov–Smirnov tests (Figure 3C). For comparing overlaps, we used chi-squared or Fisher’s exact test (Figure 2B). All analyses were carried out by R. In many cases, we combined the two types of unidirectional origin and compare them to the bidirectional origins.

## Figures and Tables

**Figure 1 ijms-24-09611-f001:**
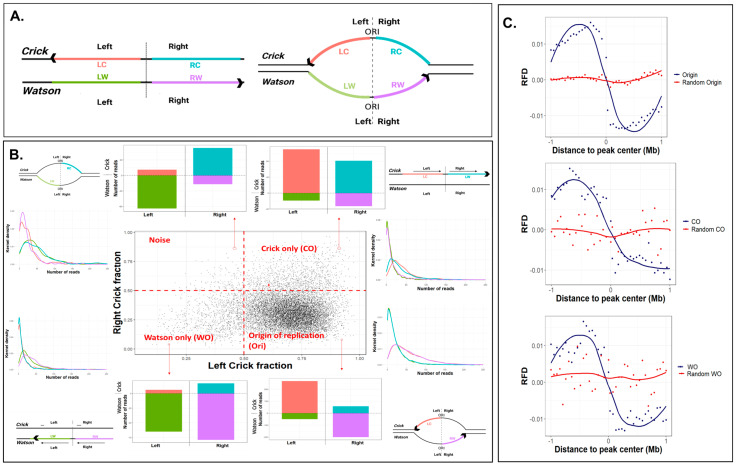
**Using ori-SSDS data for the identification of non-symmetric origins.** (**A**) Schematic representation of unidirectional replication (**left**) and bidirectional origin of replication (**right**). Arrowheads represent the leading strand. The dashed lines represent the middle of the peaks and the colors represent the right and left sides of the peaks separately for the Watson and Crick strands. (**B**) Scatterplot showing the fraction of reads mapped to the Crick strand right and left to the middle of the ori-SSDS peak in ES cells (testis data are shown in Appendix A). For each quadrant, three plots are shown: (i) a representative bar graph that shows the number of reads mapped to each strand (positive numbers for the Crick and negative for the Watson); (ii) distribution of the number of reads mapped to the Watson and Crick strands in the left and right sides of the peaks; and (iii) schematic representation of replication structure. The color codes are the same as in (**A**). (**C**) Average replication fork direction (RFD) plotted for the regions surrounding the three peak types, for real (blue) and random (red) peaks.

**Figure 2 ijms-24-09611-f002:**
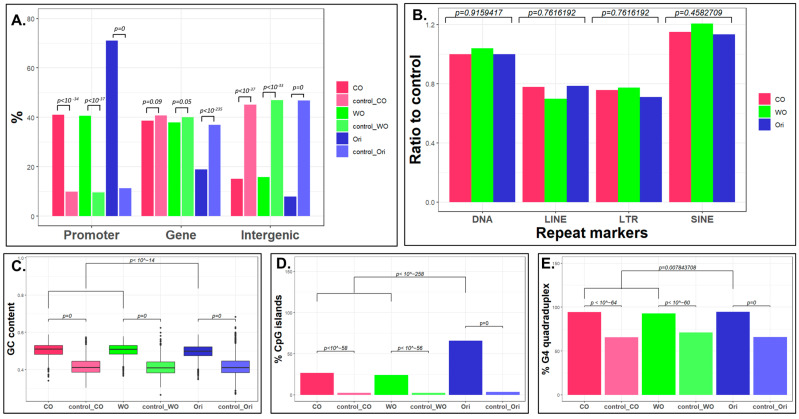
**Genetic characterization of the three types of peak in ES cells.** (**A**) Bar graph showing the percentage of each type of peak that overlaps with promoters, genes and intergenic regions in comparison to the percentage of the overlap with the random groups (see methods). The statistical significance of the differences was assessed using the chi-squared test. (**B**) Bar graph showing the difference between each type of peak and its corresponding control in the overlap with various types of repetitive sequences. Statistical significance was assessed by Fisher’s exact test. (**C**) Box plot showing the distribution of GC content in the three types of peak and their corresponding control groups. Statistical significance was assessed by a two-tailed *t*-test. (**D**) Bar graph showing the percentage of each type of peak and its corresponding control that overlaps with CpG islands. The statistical significance of the differences was assessed using the chi-squared test. (**E**) Bar graph showing the percentage of each type of peak and its corresponding control that overlaps with G4 quadruplex structures. The statistical significance of the differences was assessed using the chi-squared test. Similar results were obtained in testis (Appendix A).

**Figure 3 ijms-24-09611-f003:**
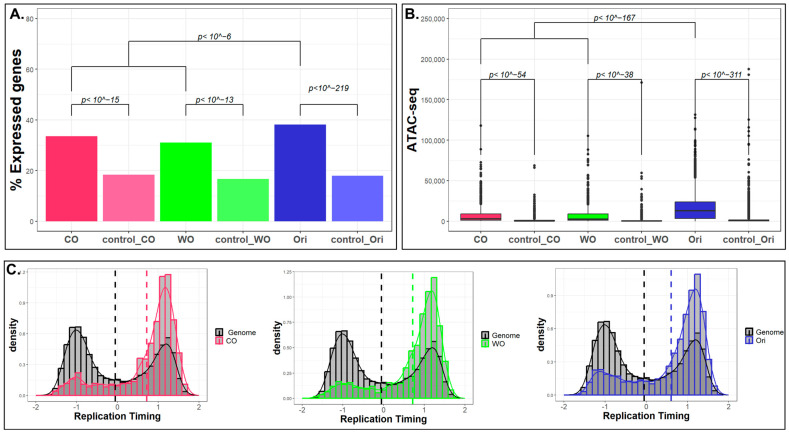
**Epigenetic characterization of the three types of peak in ES cells.** (**A**) Bar graph representation of the percentage of peaks and their corresponding controls with expressed genes (RPKM > 5). The statistical significance was assessed by chi-squared test. (**B**) Box plot showing the distribution of ATAC-seq read count in the three types of peak and their corresponding control groups. The statistical significance was assessed by two side *t*-test. (**C**) Replication timing (RT) distribution of the entire genome (black) and for the regions of the three types of peak (CO, WO and Ori in red, green and blue, respectively). All three distributions were very different from the distribution of the RT in the entire genome (*p*~0; KS test). The distribution of RT of the origins was also significantly different from that of both CO and WO (*p* < 0.002; KS test). RT distribution for testis is shown in Appendix A.

**Figure 4 ijms-24-09611-f004:**
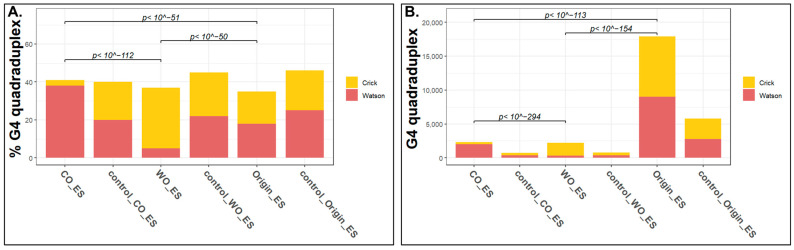
**G4 quadruplex distinguishes between the unidirectional and bidirectional origins.** (**A**) Bar graph representations of the percentage of peaks that overlap with G4 quadruplexes on either the Watson or the Crick strands for ES cells. Peaks overlapping with G4 quadruplexes on both strands were excluded from the analysis. Statistical significance was assessed by chi-squared test. (**B**) Bar graph representation of the number of G4 quadruplexes mapped to the 2 kb region surrounding each peak in a strand specific manner, both for peaks and for their corresponding controls. Statistical significance was assessed by chi-squared test. Testis data are shown in Appendix A.

**Figure 5 ijms-24-09611-f005:**
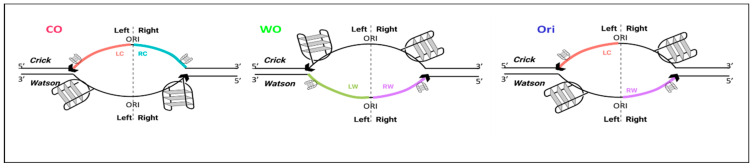
**Schematic model for the difference between unidirectional and bidirectional origins.** The leading strands are marked by arrowheads. The G4 quadruplexes are represented by small cartoons, their size representing the prevalence of those structures in each strand.

## Data Availability

No new data were created in this work.

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
