# Peer review of "Characterization of Unidirectional Replication Forks in the Mouse Genome"

_ijms, 2023, doi:10.3390/ijms24119611_

Round 1
Reviewer 1 Report
Overview
In this manuscript, the authors reanalyzed Ori-SSDS data and describe unidirectional replication forks. The unidirectional forks were defined by reads on only one strand, either Crick or Watson only. They investigated the genomic and epigenomic characteristics of unidirectional origins compared to classic origins and found only minor differences. They found enrichment of G4 sequences on the opposite strand to the leading strand of unidirectional origins, providing a possible explanation for the presence of unidirectional origins. This is an interesting computational study. My main concern is that the methodology makes a lot of assumptions and there is no experimental evidence to support the claims of unidirectional origins. Clarification of the methods and additional controls would strengthen this study.
Major points
- It’s still not clear to me that the Crick or Watson only reads are truly origins. How do the authors explain how a unidirectional fork with Crick-only or Watson-only reads would also lead to changes in fork directionality shown in 1C?
- Could the unidirectional origins defined here be due to loss of primer RNA on one strand and thus not truly unidirectional? The authors should comment on any pitfalls with their data being based entirely on one technique (Ori-SSDS).
- Is there a known mouse RFB that results in unidirectional origin/fork? If so, it would be an essential control to validate the unidirectional origins described here.
- The authors should provide clearer figures of their computational work flow for each type of experiment.
Minor points
- The P-values in figure 2A need to be fixed.
- It would be helpful if the authors provided a schematic of how Ori-SSDS works since the paper is largely based on this method.
Author Response
Point by point response
Reviewer 1
Comments and Suggestions for Authors
Overview
In this manuscript, the authors reanalyzed Ori-SSDS data and describe unidirectional replication forks. The unidirectional forks were defined by reads on only one strand, either Crick or Watson only. They investigated the genomic and epigenomic characteristics of unidirectional origins compared to classic origins and found only minor differences. They found enrichment of G4 sequences on the opposite strand to the leading strand of unidirectional origins, providing a possible explanation for the presence of unidirectional origins. This is an interesting computational study. My main concern is that the methodology makes a lot of assumptions and there is no experimental evidence to support the claims of unidirectional origins. Clarification of the methods and additional controls would strengthen this study.
Major points
- It’s still not clear to me that the Crick or Watson only reads are truly origins. How do the authors explain how a unidirectional fork with Crick-only or Watson-only reads would also lead to changes in fork directionality shown in 1C?
We thank the reviewer for raising this point, which is indeed a conundrum that requires explanation. We now clarify in the main text that the unidirectional origins are probably a consequence of transient pause in the replication that later on proceed in the same direction and this is why the OK-seq data support bidirectional replication while the Ori-SSDS data lacks a peak in one of the directions. This is now clarified in the main text (line 109-111) and further discussed in the discussion (lines 263-274; which appears already in the original manuscript). Briefly, we suggest two alternative explanations. First, it may be that the fork is blocked and the replication resumes after the block through the activity of the PrimPol polymerase. Alternatively, it may be that the paused fork resumes replication without the need of a new primer. Yet, when the replication is resumed, the SNS methodology fails to detect it since the delayed leading strand is already fused to the lagging strand that evolved to the other direction.
- Could the unidirectional origins defined here be due to loss of primer RNA on one strand and thus not truly unidirectional? The authors should comment on any pitfalls with their data being based entirely on one technique (Ori-SSDS).
We thank the reviewer for this comment and for their caution in deriving conclusions based on a single method. We have added a paragraph in the discussion section (lines 275-284), which reflects this limitation and discusses alternative interpretation of the ori-SSDS non-symmetrical peaks.
- Is there a known mouse RFB that results in unidirectional origin/fork? If so, it would be an essential control to validate the unidirectional origins described here.
This is indeed an excellent suggestion. Unfortunately, the only mammalian RFB-like cases that we are aware of are located in repetitive sequences – in telomeres (Sfeir et al., 2009) and in the rDNA (Little et al., 1993). These regions are absent from the ori-SSDS data (due to their repetitive nature) and thus cannot be used for confirmation.
- The authors should provide clearer figures of their computational work flow for each type of experiment.
Added as Figure S9.
Minor points
- The P-values in figure 2A need to be fixed.
Fixed.
- It would be helpful if the authors provided a schematic of how Ori-SSDS works since the paper is largely based on this method.
Added as Figure S1.
Reviewer 2 Report
This paper focused on the characterization of unidirectional replication forks in the mouse genome. The manuscript is overall well written.
The introduction paragraph should be incremented with more information about the topic investigated.
Please improve the references with new papers
Minor editing of English language required
Author Response
Point by point response
Reviewer 2
Comments and Suggestions for Authors
This paper focused on the characterization of unidirectional replication forks in the mouse genome. The manuscript is overall well written.
The introduction paragraph should be incremented with more information about the topic investigated.
We thank the reviewer for their suggestion and added the required information to the introduction
Please improve the references with new papers
We have added 13 additional papers to the bibliography and it now contains 15 papers that were published in the last 5 years.
Comments on the Quality of English Language
Minor editing of English language required
We have sent the manuscript to English editing by a native English speaker
